# DD-PPO: Learning Near-Perfect PointGoal Navigators from 2.5 Billion Frames

**Erik Wijmans**[1,2*] **Abhishek Kadian**[2] **Ari Morcos**[2] **Stefan Lee**[1,3] **Irfan Essa**[1]
**Devi Parikh**[1,2] **Manolis Savva**[2,4] **Dhruv Batra**[1,2]
[1]Georgia Institute of Technology   [2]Facebook AI Research
[3]Oregon State University   [4]Simon Fraser University

## Abstract

We present Decentralized Distributed Proximal Policy Optimization (DD-PPO), a method for distributed reinforcement learning in resource-intensive simulated environments. DD-PPO is distributed (uses multiple machines), decentralized (lacks a centralized server), and synchronous (no computation is ever 'stale'), making it conceptually simple and easy to implement. In our experiments on training virtual robots to navigate in Habitat-Sim (Savva et al., 2019), DD-PPO exhibits *near-linear scaling* – achieving a speedup of 107x on 128 GPUs over a serial implementation. We leverage this scaling to train an agent for *2.5 Billion* steps of experience (the equivalent of 80 years of human experience) – over *6 months of GPU-time* training in under 3 days of wall-clock time with 64 GPUs.

This massive-scale training not only sets the state of art on Habitat Autonomous Navigation Challenge 2019, but essentially 'solves' the task – *near-perfect* autonomous navigation in an *unseen* environment *without* access to a map, directly from an `RGB-D` camera and a GPS+Compass sensor. Fortuitously, error vs computation exhibits a power-law-like distribution; thus, 90% of peak performance is obtained relatively early (at 100 million steps) and relatively cheaply (under 1 day with 8 GPUs). Finally, we show that the scene understanding and navigation policies learned can be transferred to other navigation tasks – the analog of 'ImageNet pre-training + task-specific fine-tuning' for embodied AI. Our model outperforms ImageNet pre-trained CNNs on these transfer tasks and can serve as a universal resource (all models and code are publicly available).

Code: https://github.com/facebookresearch/habitat-api

Video: https://www.youtube.com/watch?v=5PBp_V5i1v4

## 1 Introduction

Recent advances in deep reinforcement learning (RL) have given rise to systems that can outperform human experts at variety of games (Silver et al., 2017; Tian et al., 2019; OpenAI, 2018). These advances, even more-so than those from supervised learning, rely on significant numbers of training samples, making them impractical without large-scale, distributed parallelization. Thus, scaling RL via multi-node distribution is of importance to AI – that is the focus of this work.

Several works have proposed systems for distributed RL (Heess et al., 2017; Liang et al., 2018a; Tian et al., 2019; Silver et al., 2016; OpenAI, 2018; Espeholt et al., 2018). These works utilize two core components: 1) workers that collect experience ('rollout workers'), and 2) a parameter server that optimizes the model. The rollout workers are then distributed across, potentially, thousands of CPUs[1]. However, synchronizing thousands of workers introduces significant overhead (the parameter server must wait for the *slowest* worker, which can be costly as the number of workers grows). To combat this, they wait for only a few rollout workers, and then *asynchronously* optimize the model.

However, this paradigm – of a single parameter server and thousands of (typically CPU) workers – appears to be fundamentally incompatible with the needs of modern computer vision and robotics communities. Over the last few years, a large number of works have proposed training virtual robots (or *'embodied agents'*) in rich 3D simulators before transferring the learned skills to reality (Beattie

---

[*]Work done while an intern at Facebook AI Research. Correspondence to etw@gatech.edu.

[1]Environments in OpenAI Gym (Brockman et al., 2016) and Atari games can be simulated on solely CPUs.

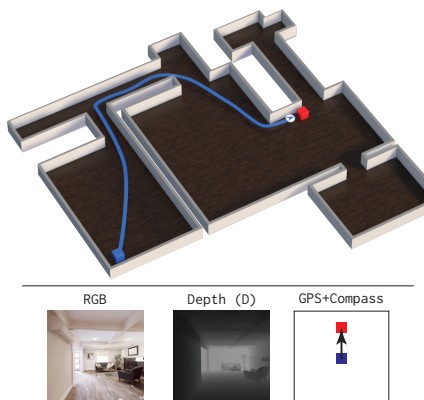
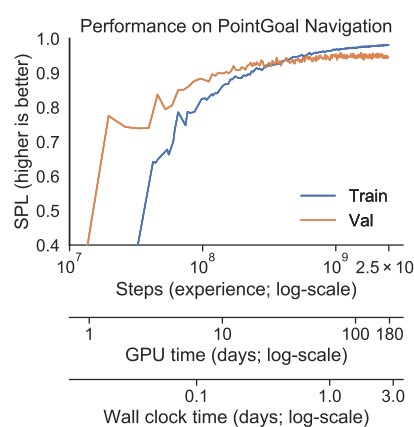

Figure 1: Left: In PointGoal Navigation, an agent must navigate from a random starting location (blue) to a target location (red) specified relative to the agent ("Go 5m north, 10m east of you") in a previously *unseen* environment *without* access to a map. Right: Performance (SPL; higher is better) of an agent equipped with RGB-D and GPS+Compass sensors on the Habitat Challenge 2019 (Savva et al., 2019) train & val sets. Using DD-PPO, we train agents for over *180 days* of GPU-time in under 3 days of wall-clock time with 64 GPUs, achieving state-of-art results and 'solving' the task.

et al., 2016; Chaplot et al., 2017; Das et al., 2018; Gordon et al., 2018; Anderson et al., 2018b; Wijmans et al., 2019; Savva et al., 2019). Unlike Gym or Atari, 3D simulators require GPU acceleration, and, consequently, the number of workers is greatly limited ($2^{5 \text{ to } 8}$ vs. $2^{12 \text{ to } 15}$). The desired agents operate from high dimensional inputs (pixels) and, consequentially, use deep networks (ResNet50) that strain the parameter server. Thus, there is a need to develop a new distributed architecture.

**Contributions.** We propose a simple, synchronous, distributed RL method that scales well. We call this method Decentralized Distributed Proximal Policy Optimization (DD-PPO) as it is decentralized (has no parameter server), distributed (runs across many different machines), and we use it to scale Proximal Policy Optimization (Schulman et al., 2017).

In DD-PPO, each worker alternates between collecting experience in a resource-intensive and GPU accelerated simulated environment and optimizing the model. This distribution is *synchronous* – there is an explicit communication stage where workers synchronize their updates to the model (the gradients). To avoid delays due to stragglers, we propose a *preemption threshold* where the experience collection of stragglers is forced to end early once a pre-specified percentage of the other workers finish collecting experience. All workers then begin optimizing the model.

We characterize the scaling of DD-PPO by the steps of experience per second with N workers relative to 1 worker. We consider two different workloads, 1) simulation time is roughly equivalent for all environments, and 2) simulation time can vary dramatically due to large differences in environment complexity. Under both workloads, we find that DD-PPO scales *near-linearly*. While we only examined our method with PPO, other on-policy RL algorithms can easily be used and we believe the method is general enough to be adapted to *off*-policy RL algorithms.

We leverage these large-scale engineering contributions to answer a key scientific question arising in embodied navigation. Mishkin et al. (2019) benchmarked classical (mapping + planning) and learning-based methods for agents with RGB-D and GPS+Compass sensors on PointGoal Navigation (Anderson et al., 2018a) (PointGoalNav), see Fig. 1, and showed that classical methods outperform learning-based. However, they trained for 'only' 5 million steps of experience. Savva et al. (2019) then scaled this training to 75 million steps and found that this trend *reverses* – learning-based outperforms classical, *even in unseen environments*! However, even with an order of magnitude more experience (75M vs 5M), they found that learning had not yet saturated. This begs the question – what are the fundamental limits of learnability in PointGoalNav? Is this task entirely learnable? We answer this question affirmatively via an 'existence proof'.

Utilizing DD-PPO, we find that agents continue to improve for a long time (Fig. 1) – not only setting the state of art in Habitat Autonomous Navigation Challenge 2019 (Savva et al., 2019), but essentially 'solving' PointGoalNav (for agents with GPS+Compass). Specifically, these agents 1) almost

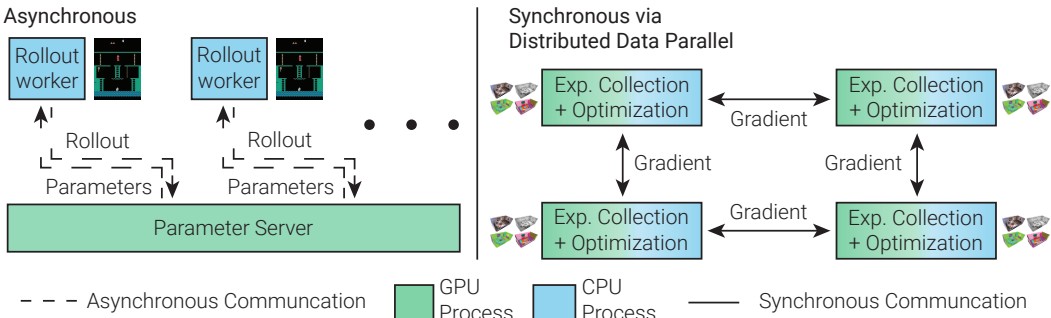

Figure 2: Comparison of asynchronous distribution (left) and synchronous distribution via distributed data parallelism (right) for RL. Left: rollout workers collect experience and asynchronously send it to the parameter-server. Right: a worker alternates between collecting experience, synchronizing gradients, and optimization. We find this highly effective in resource-intensive environments.

always reach the goal (failing on 1/1000 val episodes on average), and 2) reach it *nearly as efficiently as possible* – nearly matching (within 3% of) the performance of a *shortest-path oracle*! It is worth stressing how uncompromising that comparison is – in a *new* environment, an agent navigating without a map traverses a path nearly matching the shortest path on the map. This means there is no scope for mistakes of any kind – no wrong turn at a crossroad, no back-tracking from a dead-end, no exploration or deviation of any kind from the shortest-path. Our hypothesis is that the model learns to exploit the statistical regularities in the floor-plans of indoor environments (apartments, offices) in our datasets. The more challenging task of navigating purely from an RGB camera without GPS+Compass demonstrates progress but remains an open frontier.

Finally, we show that the scene understanding and navigation policies learned on `PointGoalNav` can be transferred to other tasks (Flee and Explore (Gordon et al., 2019)) – the analog of 'ImageNet pre-training + task-specific fine-tuning' for Embodied AI. Our models are able to rapidly learn these new tasks (outperforming ImageNet pre-trained CNNs) and can be utilized as near-perfect *neural PointGoal controllers*, a universal resource for other high-level navigation tasks (Anderson et al., 2018b; Das et al., 2018). We make code and trained models publicly available.

## 2 PRELIMINARIES: RL AND PPO

Reinforcement learning (RL) is concerned with decision making in Markov decision processes. In a partially observable MDP (POMDP), the agent receives an observation that does *not* fully specify the state ($s_t$) of the environment, $o_t$ (*e.g.* an egocentric RGB image), takes an action $a_t$, and is given a reward $r_t$. The objective is to maximize cumulative reward over an *episode*, Formally, let $\tau$ be a sequence of $(o_t, a_t, r_t)$ where $a_t \sim \pi(\cdot \mid o_t)$, and $s_{t+1} \sim \mathcal{T}(s_t, a_t)$. For a discount factor $\gamma$, which balances the trade-off between exploration and exploitation, the optimal policy, $\pi^*$, is specified by

$$\pi^* = \underset{\pi}{\operatorname{argmax}} \, \mathbb{E}_{\tau \sim \pi} \left[ R_T \right], \quad \text{where, } R_T = \sum_{t=1}^{T} \gamma^{t-1} r_t. \tag{1}$$

One technique to find $\pi^*$ is Proximal Policy Optimization (PPO) (Schulman et al., 2017), an on-policy algorithm in the policy-gradient family. Given a $\theta$-parameterized policy $\pi_\theta$ and a set of trajectories collected with it (commonly referred to as a 'rollout'), PPO updates $\pi_\theta$ as follows. Let $\hat{A}_t = R_t - \widehat{V}_t$, be the estimate of the advantage, where $R_t = \sum_{i=t}^{T} \gamma^{i-t} r_i$, and $\hat{V}_t$ is the expected value of $R_t$, and $r_t(\theta) = \frac{\pi_\theta(a_t|o_t)}{\pi_{\theta_t}(a_t|o_t)}$ be the ratio of the probability of the action $a_t$ under the current policy and the policy used to collect the rollout. The parameters are then updated by maximizing

$$\mathcal{J}^{PPO}(\theta) = E_t \left[ \min \left( \underbrace{r_t(\theta)\hat{A}_t}_{\text{importance-weighted advantage}}, \underbrace{\operatorname{clip}(r_t(\theta), 1 - \epsilon, 1 + \epsilon)\hat{A}_t}_{\text{proximity clipping term}} \right) \right] \tag{2}$$

This clipped objective keeps this ratio within $\epsilon$ and functions as a trust-region optimization method; allowing for the multiple gradient updates using the rollout, thereby improving sample efficiency.

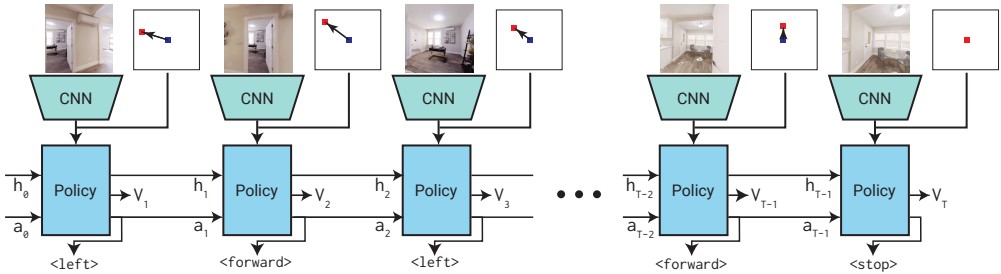

Figure 3: Our agent for `PointGoalNav`. At very time-step, the agent receives an egocentric `Depth` or `RGB` (shown here) observation, utilizes its GPS+Compass sensor to update the target position to be relative to its current position, and outputs the next action and an estimate of the value function.

## 3 DECENTRALIZED DISTRIBUTED PROXIMAL POLICY OPTIMIZATION

In reinforcement learning, the dominant paradigm for distribution is asynchronous (see Fig. 2). Asynchronous distribution is notoriously difficult – even minor errors can result in opaque crashes – and the parameter server and rollout workers necessitate separate programs.

In supervised learning, however, synchronous distributed training via data parallelism (Hillis & Steele Jr, 1986) dominates. As a general abstraction, this method implements the following: at step $k$, worker $n$ has a copy of the parameters, $\theta_n^k$, calculates the gradient, $\partial \theta_n^k$, and updates $\theta$ via

$$\theta_n^{k+1} = \texttt{ParamUpdate}\Big(\theta_n^k, \texttt{AllReduce}\big(\partial \theta_1^k, \dots, \partial \theta_N^k\big)\Big) = \texttt{ParamUpdate}\Big(\theta_n^k, \frac{1}{N}\sum_{i=1}^{N}\partial \theta_i^k\Big), \quad (3)$$

where `ParamUpdate` is any first-order optimization technique (*e.g.* gradient descent) and `AllReduce` performs a reduction (*e.g.* mean) over all copies of a variable and returns the result to all workers. Distributed DataParallel scales very well (near-linear scaling up to 32,000 GPUs (Kurth et al., 2018)), and is reasonably simple to implement (all workers synchronously running identical code).

We adapt this to on-policy RL as follows: At step $k$, a worker $n$ has a copy of the parameters $\theta_n^k$; it gathers experience (rollout) using $\pi_{\theta_n^k}$, calculates the parameter-gradients $\nabla_\theta$ via any policy-gradient method (*e.g.* PPO), synchronizes these gradients with other workers, and updates the model:

$$\theta_n^{k+1} = \texttt{ParamUpdate}\Big(\theta_n^k, \texttt{AllReduce}\big(\nabla_\theta \mathcal{J}^{PPO}(\theta_1^k), \dots, \nabla_\theta \mathcal{J}^{PPO}(\theta_N^k)\big)\Big). \quad (4)$$

A key challenge to using this method in RL is variability in experience collection run-time. In supervised learning, all gradient computations take approximately the same time. In RL, some resource-intensive environments can take significantly longer to simulate. This introduces significant synchronization overhead as every worker must wait for the slowest to finish collecting experience. To combat this, we introduce a preemption threshold where the rollout collection stage of these stragglers is preempted (forced to end early) once some percentage, $p\%$, (we find $60\%$ to work well) of the other workers are finished collecting their rollout; thereby dramatically improving scaling. We weigh all worker's contributions to the loss equally and limit the minimum number of steps before preemption to one-fourth the maximum to ensure all environments contribute to learning.

While we only examined our method with PPO, other on-policy RL algorithms can easily be used and we believe the method can be adapted to *off*-policy RL algorithms. Off-policy RL algorithms also alternate between experience collection and optimization, but differ in how experience is collected/used and the parameter update rule. Our adaptations simply add synchronization to the optimization stage and a preemption to the experience collection stage.

**Implementation.** We leverage PyTorch's (Paszke et al., 2017) `DistributedDataParallel` to synchronize gradients, and `TCPStore` – a simple distributed key-value storage – to track how many workers have finished collecting experience. See Apx. E for a detailed description with code.

## 4 EXPERIMENTAL SETUP: POINTGOAL NAVIGATION, AGENTS, SIMULATOR

**PointGoal Navigation** (`PointGoalNav`). An agent is initialized at a random starting position and orientation in a new environment and asked to navigate to target coordinates specified relative to the

agents position; no map is available and the agent must navigate using only its sensors – in our case RGB-D (or RGB) and GPS+Compass (providing current position and orientation *relative* to start).

The evaluation criteria for an episode is as follows (Anderson et al., 2018a): Let $S$ indicate 'success' (did the agent stop within 0.2 meters of the target?), $l$ be the length of the shortest path between start and target, and $p$ be the length of the agent's path, then Success weighted by (normalized inverse) Path Length SPL $= S \frac{l}{\max(l,p)}$. It is worth stressing that SPL is a highly punitive metric – to achieve SPL $= 1$, the agent (navigating without the map) must match the performance of the shortest-path oracle that has access to the map! There is no scope for any mistake – no wrong turn at a crossroad, no back-tracking from a dead-end, no exploration or deviation from the shortest path. In general, this may not even be possible in a new environment (certainly not if an adversary designs the map).

**Agent.** As in Savva et al. (2019), the agent has 4 actions, stop, which indicates the agent has reached the goal, move_forward (0.25m), turn_left (10°), and turn_right (10°). It receives 256x256 sized images and uses the GPS+Compass to compute target coordinates relative to its current state. The RGB-D agent is limited to only Depth as Savva et al. (2019) found this to perform best.

Our agent architecture (Fig. 3) has two main components – a visual encoder and a policy network.

The visual encoder is based on either ResNet (He et al., 2016) or SE (Hu et al., 2018)-ResNeXt (Xie et al., 2017) with the number of output channels at every layer reduced by half. We use a first layer of 2x2-AvgPool to reduce resolution (essentially performing low-pass filtering + down-sampling) – we find this to have no impact on performance while allowing faster training. From our initial experiments, we found it necessary to replace every BatchNorm layer (Ioffe & Szegedy, 2015) with GroupNorm (Wu & He, 2018) to account for highly correlated inputs seen in on-policy RL.

The policy is parameterized by a 2-layer LSTM with a 512-dimensional hidden state. It takes three inputs: the previous action, the target relative to the current state, and the output of the visual encoder. The LSTM's output is used to produce a softmax distribution over the action space and an estimate of the value function. See Appendix C for full details.

**Training.** We use PPO with Generalized Advantage Estimation (Schulman et al., 2015). We set the discount factor $\gamma$ to 0.99 and the GAE parameter $\tau$ to 0.95. Each worker collects (up to) 128 frames of experience from 4 agents running in parallel (all in different environments) and then performs 2 epochs of PPO with 2 mini-batches per epoch. We use Adam (Kingma & Ba, 2014) with a learning rate of $2.5 \times 10^{-4}$. Unlike popular implementations of PPO, we do not normalize advantages as we find this leads to instabilities. We use DD-PPO to train with 64 workers on 64 GPUs.

The agent receives terminal reward $r_T = 2.5$ SPL, and shaped reward $r_t(a_t, s_t) = -\Delta_{\text{geo\_dist}} - 0.01$, where $\Delta_{\text{geo\_dist}}$ is the change in geodesic distance to the goal by performing action $a_t$ in state $s_t$.

**Simulator+Datasets.** Our experiments are conducted using Habitat, a 3D simulation platform for embodied AI research (Savva et al., 2019). Habitat is a modular framework with a highly performant and stable simulator, making it an ideal framework for simulating billions of steps of experience.

We experiment with several different sources of data. First, we utilize the training data released as part of the Habitat Challenge 2019, consisting of 72 scenes from the Gibson dataset (Xia et al., 2018). We then augment this with *all* 90 scenes in the Matterport3D dataset (Chang et al., 2017) to create a larger training set (note that Matterport3D meshes tend to be larger and of better quality).[2] Furthermore, Savva et al. (2019) curated the Gibson dataset by rating every mesh reconstruction on a quality scale of 0 to 5 and then filtered all splits such that each only contains scenes with a rating of 4 or above (Gibson-4+), leaving all scenes with a lower rating previously unexplored. We examine training on the 332 scenes from the original train split with a rating of 2 or above (Gibson-2+).

## 5 BENCHMARKING: HOW DOES DD-PPO SCALE?

In this section, we examine how DD-PPO scales under two different workload regimes – homogeneous (every environment takes approximately the same amount of time to simulate) and heterogeneous (different environments can take orders of magnitude more/less time to simulate). We examine the number of steps of experience per second with N workers relative to 1 worker. We compare different values of the preemption threshold $p\%$. We benchmark training our ResNet50 PointGoalNav agent with Depth on a cluster with Nvidia V100 GPUs and NCCL2.4.7 with Infiniband interconnect.

---

[2]We use *all* Matterport3D scenes (including test and val) as we only evaluate on Gibson validation and test.

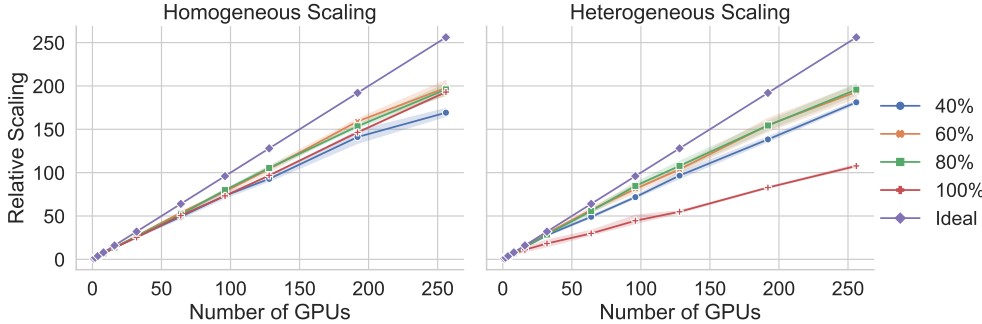

Figure 4: Scaling performance (in steps of experience per second relative to 1 GPU) of DD-PPO for various preemption threshold, $p\%$, values. Shading represents a 95% confidence interval.

**Homogeneous.** To create a homogeneous workload, we train on scenes from the Gibson dataset, which require very similar times to simulate agent steps. As shown in Fig. 4 (left), DD-PPO exhibits *near-linear scaling* (linear = ideal) for preemption thresholds larger than 50%, achieving a 196x speed up with 256 GPUs relative to 1 GPU and an 7.3x speed up with 8 GPUs relative to 1.

**Heterogeneous.** To create a heterogeneous workload, we train on scenes from both Gibson and Matterport3D. Unlike Gibson, MP3D scenes vary significantly in complexity and time to simulate – the largest contains 8GB of data while the smallest is only 135MB. DD-PPO scales poorly at a preemption threshold of 100% (no preemption) due to the substantial straggler effect (one rollout taking substantially longer than the others); see Fig. 4 (right). However, with a preemption threshold of 80% or 60%, we achieve *near-identical* scaling to the homogeneous workload! We found no degradation in performance of models trained with any of these values for the preemption threshold despite learning in large scenes occurring at a lower frequency.

## 6 MASTERING POINTGOAL NAVIGATION WITH GPS+COMPASS

In this section, we answer the following questions: 1) What are the fundamental limits of learnability in `PointGoalNav` navigation? 2) Do more training scenes improve performance? 3) Do better visual encoders improve performance? 4) Is `PointGoalNav` 'solvable' when navigating from `RGB` instead of `Depth`? 5) What are the open/unsolved problems – specifically, how does navigation without GPS+Compass perform? 6) Can agents trained for `PointGoalNav` be transferred to new tasks?

**Agents continue to improve for a long time.** Using DD-PPO, we train agents for 2.5 *Billion* steps of experience with 64 Tesla V100 GPUs in 2.75 days – 180 *GPU-days* of training, the equivalent of 80 years of human experience (assuming 1 human second per step). As a comparison, Savva et al. (2019) reached 75 million steps (an order of magnitude more than prior work) in 2.5 days using 2 GPUs – at that rate, it would take them over a month (wall-clock time) to achieve the scale of our study. Fig. 1 shows the performance of an agent with `RGB-D` and GPS+Compass sensors, utilizing an SE-ResNeXt50 visual encoder, trained on Gibson-2+ – it does not saturate before 1 billion steps[3], suggesting that previous studies were incomplete by 1-2 *orders of magnitude*. Fortuitously, error vs computation exhibits a power-law-like distribution; 90% of peak performance is obtained relatively early (100M steps) and relatively cheaply (in 0.1 day with 64 GPUs and in 1 day with 8 GPUs[4]). Also noteworthy in Fig. 1 is the strong generalization (train to val) and corresponding lack of overfitting.

**Increasing training data helps.** Tab. 1 presents results with different training datasets and visual encoders for agent with `RGB-D` and GPS+Compass. Our most basic setting (ResNet50, Gibson-4+ training) already achieves SPL of 0.922 (val), 0.917 (test), which nearly misses (by 0.003) the top of the leaderboard for the Habitat Challenge 2019 `RGB-D` track[5]. Next, we increase the size of the training data by adding in *all* Matterport3D scenes and see an improvement of ~0.03 SPL – to 0.956 (val), 0.941 (test). Next, we compare training on Gibson-4+ and Gibson-2+. Recall that Gibson-{2, 3} corresponds to poorly reconstructed scenes (see Fig. 11). A priori, it is unclear whether the net effect of this addition would be positive or negative; adding them provides diverse experience to the

---

[3]These trends are consistent across sensors (RGB), training datasets (Gibson-4+), and visual encoders.

[4]The current on-demand price of an 8-GPU AWS instance (p2.8xlarge) is $7.2/hr, or $172.8 for 1 day.

[5]https://evalai.cloudcv.org/web/challenges/challenge-page/254/leaderboard/839

| Training Dataset | Agent Visual Encoder | Validation | | Test Standard | |
|---|---|---|---|---|---|
| | | SPL | Success | SPL | Success |
| Gibson-4+ | ResNet50 | $0.922 \pm 0.004$ | $0.967 \pm 0.003$ | 0.917 | 0.970 |
| Gibson-4+ and MP3D | ResNet50 | $0.956 \pm 0.002$ | $0.996 \pm 0.002$ | 0.941 | 0.996 |
| Gibson-2+ | ResNet50 | $0.956 \pm 0.003$ | $0.994 \pm 0.002$ | 0.944 | 0.982 |
| | SE-ResNeXt50 | $0.959 \pm 0.002$ | $0.999 \pm 0.001$ | 0.943 | 0.988 |
| | SE-ResNeXt101 + 1024-d LSTM | $0.969 \pm 0.002$ | $0.997 \pm 0.001$ | 0.948 | 0.980 |

Table 1: Performance (higher is better) of different architectures for agents with RGB-D and GPS+Compass sensors on the Habitat Challenge 2019 (Savva et al., 2019) validation and test-std splits (checkpoint selected on val). 10 samples taken for each episode on val. Gibson-4+ (2+) refers to the subset of Gibson train scenes (Xia et al., 2018) with a quality rating of 4 (2) or higher. See Tab. 2 for results of the best DD-PPO agent for Blind, RGB, and RGB-D and other baselines.

agent, however, it is poor quality data. We find a potentially counter-intuitive result – adding poor 3D reconstructions to the train set improves performance on good reconstructions in val/test by ∼0.03 SPL – from 0.922 (val), 0.917 (test) to 0.956 (val), 0.944 (test). Our conjecture is that training on poor (Gibson-{2,3}) and good (4+) reconstructions leads to robustness in representations learned.

**Better visual encoders and more parameters help.** Using a better visual encoder, SE (Hu et al., 2018)-ResNeXt50 (Xie et al., 2017) instead of ResNet50, improves performance by 0.003 SPL (Tab. 1). Adding capacity to the visual encoder (SE-ResNeXt101 vs SE-ResNeXt50) and navigation policy (1024-d vs 512-d LSTM) further improves performance by 0.010 SPL.

**PointGoalNav 'solved' with RGB-D and GPS+Compass.** Our best agent – SE-ResNeXt101 + 1024-d LSTM trained on Gibson-2+ – achieves SPL of 0.969 (val), 0.948 (test), which not only sets the state of art on the Habitat Challenge 2019 RGB-D track but is also within 3-5% of the shortest-path oracle[6]. Given the challenges with achieving near-perfect SPL in new environments, it is important to dig deeper. Fig. 13 shows (a) distribution of episode lengths in val and (b) SPL vs episode length. We see that while the dataset is dominated by short episodes (2-12m), the performance of the agent is remarkably stable over long distances and average SPL is not necessarily inflated. Our hypothesis is the agent has learned to exploit the structural regularities in layouts of real indoor environments. One (admittedly imperfect) way to test this is by training a Blind agent with only a GPS+Compass sensor. Fig. 13 shows that this agent is able to handle short-range navigation (which primarily involve turning to face the target and walking straight) but performs *very poorly* on longer trajectories – SPL of 0.3 (Blind) vs 0.95 (RGB-D) at 20-25m navigation. Thus, structural regularities, in part, explain performance for short-range navigation. For long-range navigation, the RGB-D agent is extracting overwhelming signal from its Depth sensor. We repeat this analysis on two additional navigation datasets proposed by Chaplot et al. (2019) – longer episodes and 'harder' episodes (more navigation around obstacles) – and find similar trends (Fig. 14). This discussion continues in Apx. A.

**Performance with RGB is also improved.** So far we studied RGB-D as this performed best in Savva et al. (2019). We now study RGB (with SE-ResNeXt50 encoder). We found it crucial to train on Gibson-2+ and *all* of Matterport3D, ensuring diversity in both layouts (Gibson-2+) and appearance (Matterport3D), and to channel-wise normalize RGB (subtract by mean and divide by standard deviation) as our networks lack BatchNorm. Performance improves *dramatically* from 0.57 (val), 0.47 (test) SPL in Savva et al. (2019) to near-perfect success 0.991 (val), 0.977 (test) and high SPL 0.929 (val), 0.920 (test). While SPL is considerably lower than the Depth agent, (0.929 vs 0.959), interestingly, the RGB agent still reaches the goal a similar percentage of the time (99.1% vs 99.9%). This agent achieves state-of-art on the Habitat Challenge 2019 RGB track (rank 2 entry has 0.89 SPL).[5]

**No GPS+Compass remains unsolved.** Finally, we examine if we also achieve better performance on the significantly more challenging task of navigation from RGB without GPS+Compass. At 100 million steps (an amount equivalent to Savva et al. (2019)), the agent achieves 0 SPL. By training to 2.5 billion steps, we make some progress and achieve 0.15 SPL. While this is a substantial improvement, the task continues to remain an open frontier for research in embodied AI.

**Transfer Learning.** We examine transferring our agents to the following tasks (Gordon et al., 2019)

---

[6]Videos: https://www.youtube.com/watch?v=x3fk-Ylb_7s&list=UUKkMUbmP7atzznCo0LXynlA

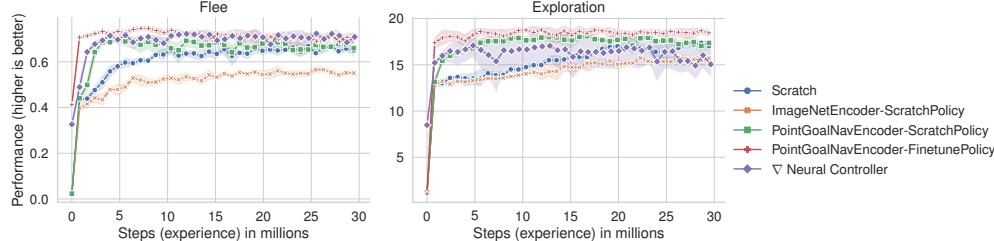

Figure 5: Performance (higher is better) on Flee (left) and Exploration (right) under five settings.

– **Flee** The agent maximizes its geodesic distance from its starting location. Let $s_t$ be the agent's position at time $t$, and $Max(s_0)$ denote the maximum distance over all reachable points, then the agent maximizes $D_T = Geo(s_T, s_0)/Max(s_0)$. The reward is $r_t = 5(D_t - D_{t-1})$.
– **Exploration** The agent maximizes the number of locations (specified by 1m cubes) visited. Let $|\text{Visited}_t|$ denote the number of location visited at time $t$, then the agent maximizes $|\text{Visited}_T|$. The reward is $r_t = 0.25(|\text{Visited}_t| - |\text{Visited}_{t-1}|)$.

We use a `PointGoalNav`-trained agent with `RGB` and GPS+Compass, remove the GPS+Compass, and transfer to these tasks under five different settings:

– **Scratch.** All parameters (visual encoder + policy) are trained from scratch for each new task. Improvements over this baseline demonstrate benefits of transfer learning.
– **ImageNetEncoder-ScratchPolicy.** The visual encoder is initialized with ImageNet pre-trained weights and frozen; the navigation policy is trained from scratch.
– **PointGoalNavEncoder-ScratchPolicy.** The visual encoder is initialized from `PointGoalNav` and frozen; the navigation policy is trained from scratch.
– **PointGoalNavEncoder-FinetunePolicy.** Both visual encoder and policy parameters are initialized from `PointGoalNav` (critic layers are reinitialized). Encoder is frozen, policy is fine-tuned.[7]
– ∇ **Neural Controller** We treat our agent as a *differentiable neural controller*, a closed-loop low-level controller than can navigate to a specified coordinate. We utilize this controller in a new task by training a light-weight high-level planner that predicts a goal-coordinate (at each time-step) for the controller to navigate to. Since the controller is fully differentiable, we can backprop through it. We freeze the controller, train the planner+controller system with PPO for the new task. The planner is a 2-layer LSTM and shares the (frozen) visual encoder with the controller.

Fig. 5 shows performance vs. experience results (higher is better). Nearly all methods outperform learning from scratch, establishing the value of transfer learning. `PointGoalNav` pre-trained visual encoders dramatically outperforms ImageNet pre-trained ones, indicating that the agent has learned generally useful scene understanding. For both tasks, fine-tuning an existing policy allows it to rapidly learn the new task, indicating that the agent has learned general navigation skills. ∇Neural Controller outperforms `PointGoalNavEncoder-ScratchPolicy` on Flee and is competitive on Exploration, indicating that the agent can indeed be 'controlled' or directed to target locations by a planner. Overall, these results demonstrate that our trained model is useful for more than just `PointGoalNav`.

## 7 RELATED WORK

**Visual Navigation.** Visual navigation in indoor environments has been the subject of many recent works (Gupta et al., 2017; Das et al., 2018; Anderson et al., 2018b; Savva et al., 2019; Mishkin et al., 2019). Our primary contribution is DD-PPO, thus we discuss other distributed works.

In the general case, computation in reinforcement learning (RL) in simulators can be broken down into 4 roles: 1) Simulation: Takes actions performed by the agent as input, simulates the new state, returns observations, reward, *etc.* 2) Inference: Takes observations as input and utilizes the agent policy to return actions, value estimate, *etc.* 3) Learner: Takes rollouts as input and computes gradients to update the policy's parameters. 4) Parameter server/master: Holds the source of truth for the policy's parameters and coordinates workers.

---

[7]Since a `PointGoalNav` policy expects a goal-coordinate, we input a 'dummy' arbitrarily-chosen vector for the transfer tasks, which the agent quickly learns to ignore.

**Synchronous RL.** Synchronous RL systems utilize a single processes to perform all four roles; this design is found in RL libraries like OpenAI Baselines (Dhariwal et al., 2017) and Py-torchRL (Kostrikov, 2018). This method is limited to a single nodes worth of GPUs.

**Synchronous Distributed RL.** The works most closely related to DD-PPO also propose to scale synchronous RL by replicating this simulation/inference/learner process across multiple GPUs and then synchronize gradients with AllReduce. Stooke & Abbeel (2018) experiment with Atari and find it not effective however. We hypothesize that this is due to a subtle difference – this distribution design relies on a single worker collecting experience from multiple environments, stepping through them in *lock step*. This introduces significant synchronization and communication costs as *every step* in the rollout must be synchronized across as many as 64 processes (possible because each environment is resource-light, *e.g.* Atari). For instance, taking 1 step in 8 parallel pong environments takes approximately the same wall-clock time as 1 pong environment, but it takes 10 times longer to take 64 steps in lock-step; thus gains from parallelization are washed out due to the lock-step synchronization. In contrast, we study resource-intensive environments, where only 2 or 4 environments per worker is possible, and find this technique to be effective. Liang et al. (2018b) mirror our findings (this distribution method can be effective for resource intensive simulation) in GPU-accelerated physics simulation, specifically MuJoCo (Todorov et al., 2012) with NVIDIA Flex. In contrast to our work, they examine scaling up to only 32 GPUs and only for homogeneous workloads. In contrast to both, we propose an adaption to mitigate the straggler effect – preempting the experience collection (rollout) of stragglers and then beginning optimization. This improves scaling for homogeneous workloads and dramatically improves scaling for heterogeneous workloads.

**Asynchronous Distributed RL.** Existing public frameworks for *asynchronous* distributed reinforcement learning (Heess et al., 2017; Liang et al., 2018a; Espeholt et al., 2018) use a single (CPU-only) process to perform the simulation and inference roles (and then replicate this process to scale). A separate process *asynchronously* performs the learner and parameter server roles (note its not clear how to use more than one these processes as it holds the source of truth for the parameters). Adapting these methods to the resource-intensive environments studied in this work (*e.g.* Habtiat (Savva et al., 2019)) encounters the following issues: 1) Limiting the inference/simulation processes to CPU-only is untenable (deep networks and need for GPU-accelerated simulation). While the inference/simulation processes could be moved to the GPU, this would be ineffective for the following: GPUs operate most efficiently with large batch sizes (each inference/simulation process would have a batch size of 1), CUDA runtime requires ~600MB of GPU memory per process, and only one CUDA kernel (function that runs on the GPU) can executed by the GPU at a time. These issue contribute and lead to low GPU utilization. In contrast, DD-PPO utilizes a single process per GPU and batches observations from multiple environments for inference. 2) The single process learner/-parameter server is limited to a single node's worth of GPUs. While this not a limitation for small networks and low dimensional inputs, our agents take high dimensional inputs (*e.g.* a Depth sensor) and utilize large neural networks (ResNet50), thereby requiring considerable computation to compute gradients. In contrast, DD-PPO has no parameter server and every GPU computes gradients, supporting even very large networks (SE-ResNeXt101).

**Straggler Effect Mitigation.** In supervised learning, the straggler effect is commonly caused by heterogeneous hardware or hardware failures. Chen et al. (2016) propose a pool of $b$ "back-up" workers (there are $N + b$ workers total) and perform the parameter update once $N$ workers finish. In comparison, their method a) requires a parameter server, and b) discards all work done by the stragglers. Chen et al. (2018) propose to dynamically adjust the batch size of each worker such that all workers perform their forward and backward pass in the same amount of time. Our method aims to reduce variance in *experience collection* times. DD-PPO dynamically adjusts a worker's batch size as a necessary side-effect of preempting experience collection in on-policy RL.

**Distributed Synchronous SGD.** Data parallelism is a common paradigm in high performance computing (Hillis & Steele Jr, 1986). In this paradigm, parallelism is achieved by workers performing the *same* work on *different* data. This paradigm can be naturally adapted to supervised deep learning (Chen et al., 2016). Works have used this to achieve state-of-the-art results in tasks ranging from computer vision (Goyal et al., 2017; He et al., 2017) to natural language processing (Peters et al., 2018; Devlin et al., 2018; Ott et al., 2019). Furthermore, multiple deep learning frameworks provide simple-to-use wrappers supporting this parallelism model (Paszke et al., 2017; Abadi et al., 2015; Sergeev & Balso, 2018). We adapt this framework to reinforcement learning.

## 8 ACKNOWLEDGEMENTS

The Georgia Tech effort was supported in part by NSF, AFRL, DARPA, ONR YIPs, ARO PECASE. The views and conclusions contained herein are those of the authors and should not be interpreted as necessarily representing the official policies or endorsements, either expressed or implied, of the U.S. Government, or any sponsor.

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

| Geodesic Distance (rows) / SPL (cols) | 0.00-0.20 | 0.20-0.50 | 0.50-0.90 | 0.90-0.95 | 0.95-1.00 |
|---|---|---|---|---|---|
| 10.0-11.7 | | | | | |
| 11.7-13.4 | | | | | |
| 13.4-15.0 | | | | | |
| 15.0-16.7 | | | | | |
| 16.7-26.8 | | | | | |

Figure 6: Example episodes broken down by geodesic distance between agent's spawn location and target (on rows) vs SPL achieved by the agent (on cols). Gray represents navigable regions on the map while white is non-navigable. The agent begins at the blue square and navigates to the red square. The green line shows the shortest path on the map (or oracle navigation). The blue line shows the agent's trajectory. The color of the agent's trajectory changes changes from dark to light over time. Navigation dataset from the longer validation episodes proposed in Chaplot et al. (2019).

## A    ADDITIONAL ANALYSIS AND DISCUSSION

In this section, we continue the analysis of our agent and examine differences in its behavior from a classical, hand-designed agent – the map-and-plan baseline agent proposed in Gupta et al. (2017).

**Intricacies of SPL.** Given an agent that always reaches the goal ($\approx$100% success), SPL can be seen as measuring the efficiency of an agent vs. an oracle – *i.e.* an SPL of 0.95 means the agent is 5% less efficient than an oracle. Given the challenges of near-perfect autonomous navigation without a map in novel environments we outlined, being 5% less efficient than an oracle seems near-impossible. However, this comparison/view is potentially miss-leading. Percentage errors are potentially miss-leading for long paths. Over a 10 meter episode, the agent can deviate from the oracle path by up-to a meter and still be within 10%. As a consequence, significant qualitative errors can result in an insignificant quantitative error (see Fig. 6).

**Error recovery.** Given the near-perfect performance of our agent (on average), we explicitly examine if it is able to recover

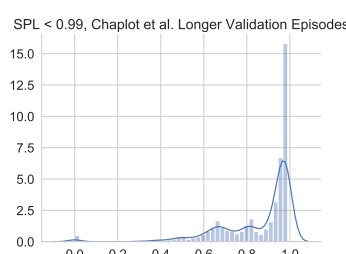

Figure 7: Histogram of SPL for non-perfect (SPL<0.99) episodes.

from its own navigation errors. Fig. 6 column 3 shows several examples of error recovery, including several well executed backtracks (video: https://www.youtube.com/watch?v=a8AugVLSJ50), indicating that the agent is effective at recovering from its own navigation errors. Next, we look at the statistics of non-perfect (SPL<0.99) episodes on the longer validation episodes proposed in Chaplot et al. (2019). Non-perfect episodes make up the majority of episodes (54%, see Fig. 7) with an average SPL of 0.85 (99.0% success) – compared to 0.92 SPL (99.5% success) over all episodes. Thus there are many episodes where the agent makes significant deviation from the shortest path and reaches the goal (a 15% deviation on long trajectories (>10m) is significant).

**When does the agent fail?** Column 2 in Fig. 6 shows that the agent performs poorly when the ratio of the geodesic distance to goal and euclidean distance to goal. However, the agent is able to eventually overcome this failure mode and reach the goal in most cases.

Row 1 column 1 in Fig. 6 shows that the agent fails or performs poorly when it needs to go slightly up/down stairs. The data-set generation process used in Savva et al. (2019) only guarantees a start and goal pair won't be on *different* floors, but there remains a possibility that the agent will need to traverse the stairs slightly. However, these situations are rare, and, in general, the stairs should be avoided. Furthermore, the GPS sensor provides location in 2D, not 3D.

The remaining failure cases of column 1 in Fig. 6 show that a singular location in one environment acts as a sink for the agent (once it enters this location, it is almost never able to leave it). At this location, there is a large hole in the mesh (an entire wall is missing). Utilizing visual encoders that explicitly handle missing values may allow the agent to overcome this failure mode.

**Differences from a classical agent.** We compare the *behavior* of our agent with the classical map-and-plan baseline agent proposed in Gupta et al. (2017). This agent achieves 0.92 val (0.89 test) SPL with 0.976 success.[8] By comparing and contrasting qualitative behaviors, we can determine what behaviors learning-based methods enable. We make the following observation.

The learned agent is able to recover from unexpected collisions without hurting SPL. The map-and-plan baseline agent incorporates a specific collision recovery behavior where, after repeated collisions, the agent turns around and backs up 1.25m. This behavior brings the obstacle into view, maps it, and then allows the agent to create a plan to avoid it. In contrast, our agent is able to navigate around unseen obstacles without such a large impact on SPL. Determining the set of action sequences and heuristics necessary to do this is what learning enables.

## B    RELATED WORK CONTINUED

**Straggler Effect Mitigation.** In supervised learning, the straggler effect is commonly caused by heterogeneous hardware or hardware failures. Chen et al. (2016) propose a pool of $b$ "back-up" workers (there are $N + b$ workers total) and perform the parameter update once $N$ workers finish. In comparison, their method a) requires a parameter server, and b) discards all work done by the stragglers. Chen et al. (2018) propose to dynamically adjust the batch size of each worker such that all workers perform their forward and backward pass in the same amount of time. Our method aims to reduce variance in *experience collection* times. DD-PPO dynamically adjusts a worker's batch size as a necessary side-effect of preempting experience collection in on-policy RL.

**Distributed Synchronous SGD.** Data parallelism is a common paradigm in high performance computing (Hillis & Steele Jr, 1986). In this paradigm, parallelism is achieved by workers performing the *same* work on *different* data. This paradigm can be naturally adapted to supervised deep learning (Chen et al., 2016). Works have used this to achieve state-of-the-art results in tasks ranging from computer vision (Goyal et al., 2017; He et al., 2017) to natural language processing (Peters et al., 2018; Devlin et al., 2018; Ott et al., 2019). Furthermore, multiple deep learning frameworks provide simple-to-use wrappers supporting this parallelism model (Paszke et al., 2017; Abadi et al., 2015; Sergeev & Balso, 2018). We adapt this framework to reinforcement learning.

## C    AGENT DESIGN

In this section, we outline the exact agent design we use. We break the agent into three components: a visual encoder, a goal encoder, and a navigation policy.

**Visual Encoder.** Out visual encoder uses one of three different backbones, ResNet50 (He et al., 2016), Squeeze-Excite(SE) (Hu et al., 2018)-ResNeXt50 (Xie et al., 2017), and SE-ResNeXt101. For all backbones, we reduce the number of output channels at each layer by half. We also add a `2x2-AvgPool` before each backbone so that the effective resolution is 128x128. Given these modifications, each backbone produces a 1024x4x4 feature map. We then convert this to a 128x4x4 feature map with a `3x3-Conv`.

We replace every BatchNorm layer with GroupNorm (Wu & He, 2018) to account for the highly correlated trajectories seen in on-policy RL and massively distributed training.

---

[8] https://github.com/s-gupta/map-plan-baseline#results

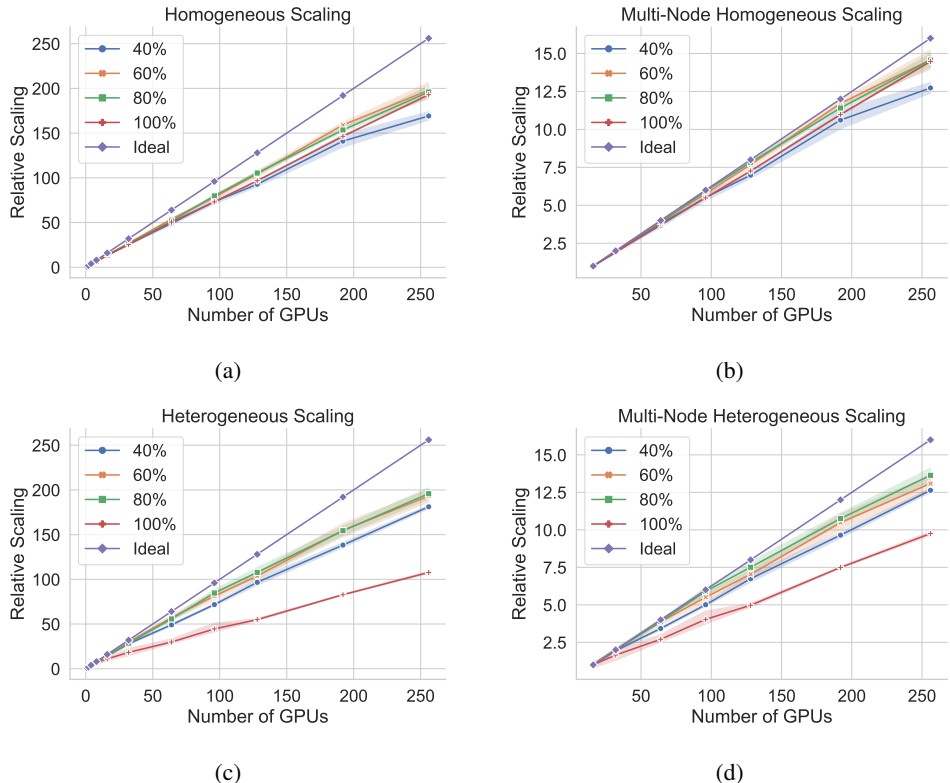

Figure 8: Scaling of DD-PPO under homogeneous and heterogeneous workloads for various different values of the percentage of rollouts that are fully completed by optimizing the model. Shading represents a bootstrapped 95% confidence interval.

**Goal encoder.** Habitat (Savva et al., 2019) provides the vector pointing to the goal in ego-centric polar coordinates. We convert this to magnitude and a unit vector, *i.e.* [d, $\theta$] to [d, $\cos(\theta)$, $\sin(\theta)$], to account for the discontinuity at the $x$-axis in polar coordinates. We pass the goal vector to a fully connected layer, resulting in a 32-dimensional representation.

**Navigation Policy.** Our navigation policy takes the 64x4x4 feature map from the visual encoder, flattens it, and then converts the 2048-d vector to the same size as the hidden size via a fully-connected layer. It then concatenates this vector with output of the goal encoder, and a 32-dimensional embedding of the previous action taken (or the start-token in the case of the first action) and then passes this to a 2-layer LSTM with either a 512-dimensional or 1024-dimensional hidden dimension. The output of the LSTM is used as input to a fully connected layer, resulting in a soft-max distribution of the action space and an estimate of the value function.

## D    ADDITIONAL SCALING DETAILS

We use the following procedure for benchmarking the throughput of our proposed DD-PPO: Each optimizer selects 4 scenes at random and then performs the process of collecting experience and optimizing the model based on that experience 10 times. We calculate throughput as the total number of steps of experience collected over the last 5 rollout/optimizing steps divided by the amount of time taken. We repeat this procedure over 10 different random seeds (we use the same random seeds for all variations of number of GPUs and sync-fraction values).

## E    DD-PPO IMPLEMENTATION

Utilizing `Distributed Data Parallel` in supervised learning is straightforward as frameworks such as PyTorch (Paszke et al., 2017) provide a simple wrapper. The recommended way to use these wrappers is to first write training code that runs on a single GPU and then enable distributed training via the wrapper. We follow a similar approach. Given an implementation of

| Perception | Method | Validation | | Test Standard | |
|---|---|---|---|---|---|
| | | SPL | Success | SPL | Success |
| Blind | Random | 0.02 | 0.03 | 0.02 | – |
| | Forward-only | 0.00 | 0.00 | 0.00 | – |
| | Goal-follower | 0.23 | 0.23 | 0.23 | – |
| | DD-PPO (RL) | $0.729 \pm 0.005$ | $0.973 \pm 0.003$ | 0.676 | 0.947 |
| RGB | DD-PPO (RL) | $0.929 \pm 0.003$ | $0.991 \pm 0.002$ | 0.920 | 0.977 |
| RGB-D (Depth) | DD-PPO (RL) | $0.969 \pm 0.002$ | $0.997 \pm 0.001$ | 0.948 | 0.980 |

Table 2: Performance (higher is better) of various sensors and agent methods on the Habitat Challenge 2019 (Savva et al., 2019) validation and test splits (checkpoint selected on val). Random, Forward-only, and Goal-follower taken from Savva et al. (2019). Best visual encoder reported for DD-PPO.

PPO that runs on one GPU we create a decentralized distributed variant by adding gradient synchronization, leveraging highly performant code written for this purpose in popular deep-learning frameworks, *e.g.* tf.distribute.MirroredStrategy in TensorFlow (Abadi et al., 2015) and torch.nn.parallel.DistributedDataParallel in PyTorch. Note that care must be taken to synchronize any training or rollout statistics between workers – in most cases these can also be synchronized via AllReduce.

We track how many workers have finished the experience collection stage with a distributed key-value storage – we use PyTorch's torch.distributed.TCPStore, however almost any distributed key-value storage would be sufficient.

See Fig. 9 for an example implementation which adds 1) gradient synchronization via torch.nn.parallel.DistributedDataParallel, and 2) preempts stragglers by tracking the number of workers have finished the experience collection stage with a torch.distributed.TCPStore.

See Fig. 10 for a visual depiction of DD-PPO.

## F    TRANSFER EXPERIMENTS ADDITIONAL DETAILS

For the transfer learning experiments, we utilize the same PPO hyper-parameters as the PointGoalNav experiments. We use DD-PPO to train with 8 workers on 8 GPUs. We train our agents on Gibson-4+ and evaluate on the Habitat Challenge 2019 Validation scene and *starting* locations (the goal location is simply discarded).

The ImageNet encoder is trained using the same hyper-parameters and training procedure as Xie et al. (2017) with no data-augmentation.

## G    NEURAL CONTROLLER ADDITIONAL DETAILS

The planner for neural controller used in Sec. 6 shares the same architecture as our agent's policy, but utilizes a 512-d hidden state. It takes as input the previous action of the controller (or the start token), and the output of the visual encoder (which is shared with the controller). The output of the LSTM is then used to produced an estimate of the value function and a 3-dimensional vector specifying the PointGoal in magnitude and unit direction vector format. The magnitude competent is passed through an ELU activation and offset by 0.75. Each component of the unit direction vector is passed through a tanh activation – note that we do not re-normalize this vector have a length of 1 as we find doing so both unnecessary and harder to optimize.

```python
master_addr = # <hostname of world rank 0's machine>
master_port = # <free TCP port on world rank 0's machine>
world_rank = # <worker's unique ID>
world_size = # <number of workers>
local_rank = # <the ID of the GPU to use>

# Setup the group of workers
store = torch.distributed.TCPStore(
    master_addr,
    master_port,
    world_size,
    world_rank == 0,
)

torch.distributed.init_process_group(
    backend="NCCL",
    world_size=world_size,
    rank=world_rank,
    store=store,
)

# Tracks how many workers have finished their rollout
num_workers_done = torch.distributed.PrefixStore(
    "num_workers_done", store
)

device = torch.device("cuda", local_rank)
model = PolicyNetwork(...)
model.to(device)
# Add gradient synchronization to the model
model = torch.nn.parallel.DistributedDataParallel(
    model, [device], device
)

while not_converged():
    num_workers_done.set("done", "0")
    for step in range(max_experience_steps):
        collect_step(model)
        # Preempt stragglers
        if (
            int(num_workers_done.get("done"))
            > preemption_threshold * world_size
            and step >= max_experience_steps / 4
        ):
            break
    # Mark that a worker is done collecting experience
    num_workers_done.add("done", 1)

    # Update the model using PPO
    for _ in range(n_ppo_epochs):
        for _ in range(n_ppo_batch):
            batch = get_batch()
            loss = evaluate(model, batch)
            loss.backward()
            # DistributedDataParallel automatically
            # performs an AllReduce on all gradients
            # during the backward call.
            # If this wasn't being used, here is where
            # calls to AllReduce on the gradients would
            # be made.
            step_optimizer(model)
```

Figure 9: Implementation of DD-PPO using PyTorch (Paszke et al., 2017) v1.1 and the NCCL backend. We use SLURM to populate the world_rank, world_size, and local_rank fields.

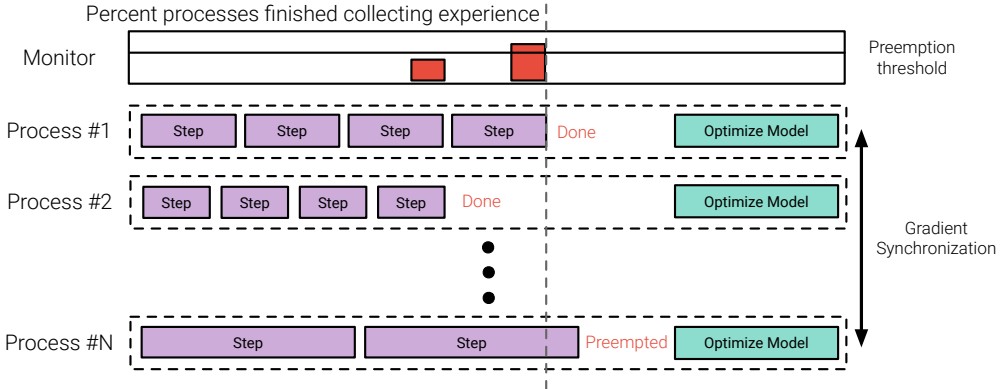

Figure 10: Illustration of DD-PPO. Processes collecting experience in environments that are more costly to simulate (stragglers) have their experience collection stage preempted such that other processes do not have to wait for them. Note that we implement the monitor with a simple key-value storage and have processes preempt themselves. Note that the order of processes is irrelevant and done solely for aesthetic purposes.

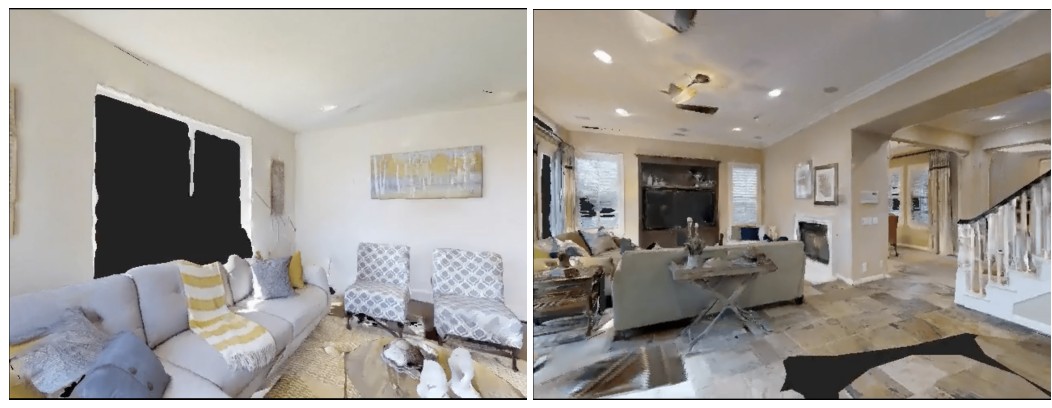

2: big holes or significant texture issues, but good reconstruction

3: small holes, some texture issues, good reconstruction

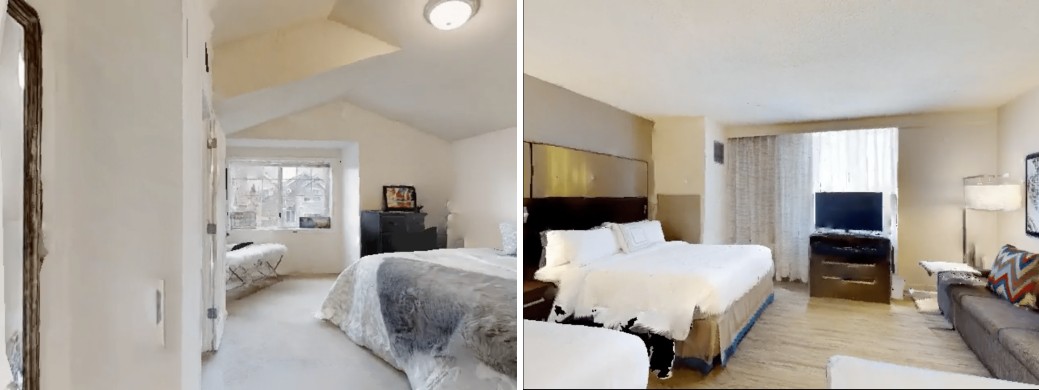

4: no holes, some texture issues, good reconstruction    5: no holes, uniform textures, good reconstruction

Figure 11: Examples of Gibson meshes for a given quality rating from Savva et al. (2019)

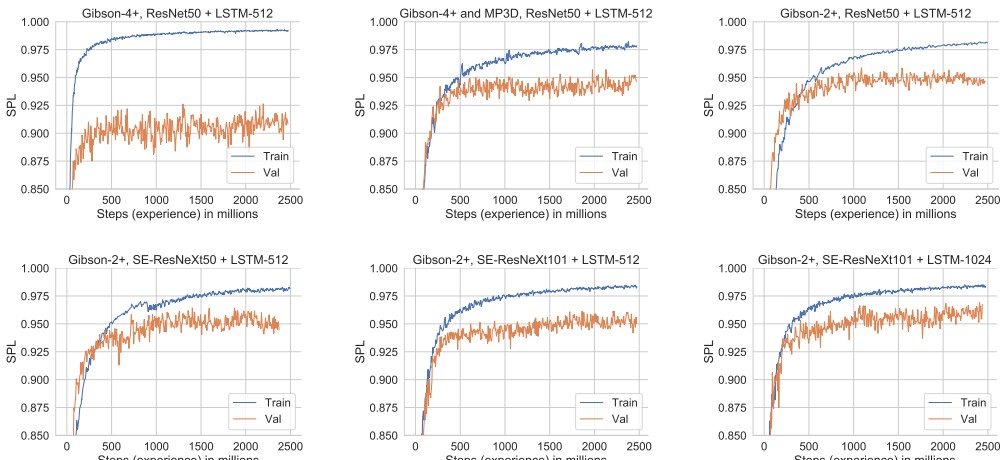

Figure 12: Training and validation performance (in SPL; higher is better) of different architectures for `Depth` agents with GPS+Compass on the Habitat Challenge 2019 (Savva et al., 2019). Gibson (Xia et al., 2018)-4+ refers to the subset of Gibson train scenes with a quality rating of 4 or better. Gibson-4+ and MP3D refers to training on both Gibson-4+ and *all* of Matterport3D. Gibson-2+ refers to training on the subset of Gibson train scenes with a quality rating of 2 or better.

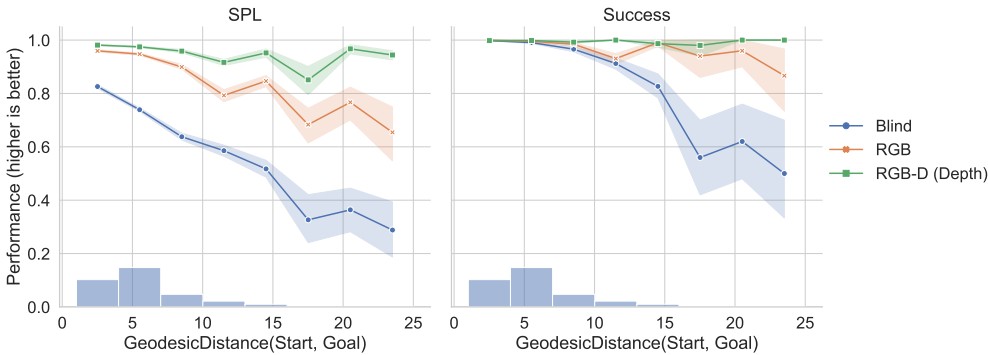

Figure 13: Performance vs. Geodesic Distance from start to goal for `Blind`, `RGB`, and `RGB-D` (using `Depth` only) models trained with DD-PPO on the Habitat Challenge 2019 (Savva et al., 2019) validation split. Bars at the bottom represent the fraction of episodes within each geodesic distance bin.

Figure 14: Performance vs. Geodesic Distance from start to goal for `Blind`, `RGB`, and `RGB-D` (using `Depth` only) models trained with DD-PPO on the longer and harder validation episodes proposed in Chaplot et al. (2019). Bars at the bottom represent the fraction of episodes within each geodesic distance bin.

