# OpenReview forum: "DD-PPO: Learning Near-Perfect PointGoal Navigators from 2.5 Billion Frames"
_ICLR.cc/2020/Conference — Accept (Poster)_

### Official Review · AnonReviewer2 · 2019-10-21
**Official Blind Review #2**

**Rating:** 8

**Review:**

The contribution of this paper is twofold. First the authors propose and implement and new scalable RL training procedure they call 'Decentralized Distributed Proximal Policy Optimization' (DD-PPO). This method extends PPO in a way that is 'synchronous' and 'distributed': no parameter server exists to asynchronously collect experience from worker agents and instead an 'explicit communication state' exists during which all workers communicate gradient updates between one another to update parameters based on their experiences. However, naively implementing this approach is limited by 'stragglers', the slowest of the workers that all other agents need to wait for. To overcome this limitation, the authors add a 'preemption threshold' which halts worker rollouts after a certain percentage has completed. The authors use this procedure to tackle the task of PointGoal navigation, in which an agent tries to reach a point in space specified by its relative location to the agent. They achieve state-of-the-art performance (arguably 'mastery') on this task. Finally, the authors show that their learned policy transfers reasonably well to two other navigation tasks, 'flee' and 'exploration'. Code is also provided.

This is an impressive paper. The algorithmic contribution is useful and the results are state-of-the-art. The paper is written very clearly, and it was a pleasure to read. The investigation of the task of PointGoal Navigation, and the impact of various changes on the overall performance of the system . The transfer tasks, to the 'flee' and 'exploration' tasks, were interesting, and further reinforced the results in the paper.

My only high-level 'complaint' is that the title might be tempered: 'Mastering PointGoal Navigation' is a very strong claim. The results in the paper are very impressive, but 'Mastering', in my mind, implies (among other things) that the agent is guaranteed to reach its goal, though I understand that it is perhaps a minor point, considering the failure rate is systematically <3%. I also make a comment later about how the Matterport 3D environments may be a biased set, and that there are almost certainly environments that *would* make this system fail, for one reason or another. I am willing to accept the title as-is (and could be overwritten by other reviewers), but I would like some discussion on a more appropriate tile, if possible.

I recommend this paper for acceptance.

Minor comments:
- Introduction: "Thus, there is a need to develop a different distributed architecture." It is unclear how this sentence logically follows from the previous sentence. I am on board with the vision in general, and it is quite clear to me: having a single parameter server is limiting. However, that the move from CPU- to GPU-based agents is what reveals this limitation is not quite so clear. Reworking this paragraph (or perhaps adding a sentence) to make this clearer would be helpful.
- Introduction: "This means there is no scope for mistakes of any kind --- no wrong turn at a crossroad, ..." This is a fascinating point, though I feel reflects a bias in the types of environments that appear in the training data set. It is a defining behavior of intelligent embodied agents that they are capable of recovering from their mistakes; in general, real-world environments and navigation tasks have inherent ambiguity that cannot be resolved without exploration (a 'maze' is an extreme example of this). One question comes to mind: if the environments were ambiguous in this way, *could* the agent recover from its mistakes. There is not much evidence in the paper one way or the other. Since there are few examples in the data that seem to bring about such scenarios, it is not obvious how well this agent would perform in the face of this uncertainty. However, the results later in the paper about how "No GPS+Compass remains unsolved" is related to this point. A comment (or a figure in the Appendix) from the authors describing some of the failure cases would be instructive: do these failures occur in environments with such an ambiguity or, conversely, are there some examples in which the agent succeeds in reaching the goal but must overcome failure during its travel?
- The acronym 'SL' (for Supervised Learning) is not defined and is used only once. Prefer using 'supervised learning'.

**Experience Assessment:**

I have published one or two papers in this area.

**Review Assessment: Checking Correctness Of Derivations And Theory:**

N/A

**Review Assessment: Checking Correctness Of Experiments:**

I assessed the sensibility of the experiments.

**Review Assessment: Thoroughness In Paper Reading:**

I read the paper at least twice and used my best judgement in assessing the paper.

---

> ### Author Response · Authors · 2019-11-14
> **Response to R2**
>
> Thank you for the detailed review!  We are thrilled that you found that “the paper is written very clearly, and it was a pleasure to read”.
>
> > the title might be tempered
>
> We see your point.  Our intent was for the title to only apply to the setting we study in this paper (PointGoal Navigation with GPS+Compass, no noise, and in realistic environments), but we see how this may be unclear.  We have changed the title to “DD-PPO: Learning Near-Perfect PointGoal Navigators from 2.5 Billion Frames”.  If the reviewer would like us to qualify or temper this down further, we will be happy to do so; please let us know.
>
> > Introduction: "Thus, there is a need to develop a different distributed architecture." ...
>
> The high-dimensional inputs (observations in pixel space) and size of networks (ResNet50) commonly used in these rich GPU-accelerated environments reveals this limitation.  Current public async RL frameworks utilize the parameter server to perform the parameter update as well as compute gradients, limiting the gradient computation to a single machine’s worth GPUs.  We are able to distribute not only experience collection, but the gradient calculation across multi-machines, allowing us to train very deep and very large networks with RL from scratch.  We have clarified this point in the paper.
>
> >  The acronym 'SL' (for Supervised Learning) is not defined and is used only once. Prefer using 'supervised learning'.
>
>
> We have updated the paper.
>
>
> >  Introduction: "This means there is no scope for mistakes of any kind --- no wrong turn at a crossroad, ..." This is a fascinating point, though...
>
> Thank you for this point, it sparked a long discussion among the authors and other collaborators in our lab.  We have updated the paper to provide additional analysis on the agents ability to recover from its own navigation error (the agent commonly recovers from small errors and back-tracks well).  We have also added additional analysis on the agent’s failure modes and a qualitative comparison with a strong classical agent.  We hope these will shed some more light on our results. Thank you again for the suggestion.

---

### Official Review · AnonReviewer3 · 2019-10-22
**Official Blind Review #3**

**Rating:** 8

**Review:**

The paper presents a novel scheme of distributing PPO reinforcement learning algorithm for hundreds of GPUs. Proposed technique was validated for pointgoal visual navigation task on recently introduced Habitat challenge and sim.

Besides the technical contribution, paper shows that when have enough computational power of billions simulation runs, it is possible to learn nearly perfect visual navigation (given RGBD + GPS inputs) via reinforcement learning.
Authors also study the task itself and show that it is yet not possible to achieve a good results without dense (each step) GPS signal, while the "Blind" agent, which has only GPS+compass error achieves quite high results given the billion-scale training time.
This suggests that PointGoal navigation with dense GPS signal is might be a poor choice to benchmark RL algorithms and we should proceed to harder tasks.


Overall I like the paper a lot and think that it should be accepted.

***
I haven`t changed my mind after the rebuttal: the paper is good and should be accepted.

**Experience Assessment:**

I have published one or two papers in this area.

**Review Assessment: Checking Correctness Of Derivations And Theory:**

I did not assess the derivations or theory.

**Review Assessment: Checking Correctness Of Experiments:**

I carefully checked the experiments.

**Review Assessment: Thoroughness In Paper Reading:**

I read the paper thoroughly.

---

> ### Author Response · Authors · 2019-11-14
> **Response to R3**
>
> > This suggests that PointGoal navigation with dense GPS signal is might be a poor choice to benchmark RL algorithms and we should proceed to harder tasks.
>
> Agreed.  We hope that our algorithm, DD-PPO, and our pretrained models will help accelerate progress on harder tasks like PointGoal Navigation without GPS+Compass, ObjectGoal/RoomGoal Navigation, Instruction Following, etc.

---

### Official Review · AnonReviewer1 · 2019-10-25
**Official Blind Review #2**

**Rating:** 3

**Review:**

Summary: This paper proposes a Decentralized Distributed architecture for PPO. The idea was demonstrated on Habitat-Sim, the Habitat Autonomous Navigation Challenge 2019. The experiment shows how the implementation can achieve a speedup in training and near-linear scalability, over non-distributed and centralized architecture.

Overall, the paper is well written and easy to follow. The idea is somewhat technical, e.g. distributed implementation of PPO and preemption for the rollouts of stragglers. The experimental results look promising but lack extensive evaluations given such a technical contribution. Therefore the paper has limited contributions. I have some following major comments.

1. As it is a technical paper, it would be nice if the authors could describe more on the implementation of challenges and the hacks.

2. The experiments could also be compared to one or two other distributed RL frameworks. It would also be helpful if detailed experiment settings are detailed, e.g. GPU characteristics, DDPPO's hyperparameters, etc.

3. The Transfer Learning tasks is a general setting for for any methods, which are not limited to only Distributed Approach. The results and setting there do not have links such as why and how distributed approaches help such transfer learning.


* minor comments:
- Eq.1: expectation should also be w.r.t transition's stochasticity.

**Experience Assessment:**

I have read many papers in this area.

**Review Assessment: Checking Correctness Of Derivations And Theory:**

I assessed the sensibility of the derivations and theory.

**Review Assessment: Checking Correctness Of Experiments:**

I assessed the sensibility of the experiments.

**Review Assessment: Thoroughness In Paper Reading:**

I read the paper at least twice and used my best judgement in assessing the paper.

---

> ### Author Response · Authors · 2019-11-14
> **Response to R1 (part 1 of 2)**
>
> >  As it is a technical paper, it would be nice if the authors could describe more on the implementation of challenges and the hacks.
>
> As noted on page 4 under the “Implementation” heading, we provide additional implementation details in the appendix (section D, DD-PPO implementation details, page 12) and a python implementation of DD-PPO built with primitives from a PPO implementation and PyTorch (Figure 7, page 14).  As stated at the end of the abstract and restated at the end of the introduction, we have committed to making this code and pre-trained models publicly available. For full transparency, a “to-be public” implementation is available with this submission (repasting link: https://drive.google.com/open?id=1J8oytPjKGCIIArGh9JtDZ98s8uAKsMbs). Since submission, our collaborators have used the “to-be public” implementation to reproduce our results and scale their own research.   Furthermore, we are in the process of adding this implementation to Habitat-API RL codebase so that it is available to all users of Habitat. If there are any specific questions about our implementation, we are happy to answer them.
>
> >  The experiments could also be compared to one or two other distributed RL frameworks.
>
> As described in paragraphs 2 and 3 of the introduction (page 1), existing RL frameworks are incompatible with the computational requirements of 3D simulated environments such as Habitat (studied in this paper).
>
> Consider RLLib (Liang et al., 2018, https://ray.readthedocs.io/en/latest/rllib.html). It assumes cheap-to-simulate environments (e.g. CartPole-V0 or Pong), CPU-only inference for the agent (e.g. fully-connected + LSTM-64 + fully-connected), and 32 to 64 CPU-based rollout workers per GPU-based optimizer. All three of these assumptions are violated for the problem setting in our paper. 3D simulators (e.g. Habitat, Gibson, AI2 THOR) effectively require GPU acceleration. Our agent has hundreds of layers and tens of millions of parameters (SE-ResNeXt101 + LSTM-1024). And because of GPU vs. CPU price differences, the number of rollout workers is greatly limited ($2^{5 \text{~to~} 8}$ for GPU vs. $2^{12 \text{~to~} 15}$ for CPU).
>
> For all of these design differences, existing frameworks like RLLib simply do not scale to our setting. Specifically, even if we ran 3D simulators on CPUs, the implementation would be so slow that reaching millions of frames of experience would take decades of CPU-time, and we need to reach billions for the experiments in this paper. We have expanded our comparison to asynchronous distributed RL methods to clarify this in the draft.
>
> If the reviewer has any specific recommendations of systems to try (that are publicly available), we will be happy to run those experiments.
>
> >  It would also be helpful if detailed experiment settings are detailed, e.g. GPU characteristics, DDPPO's hyperparameters, etc.
>
> GPU characteristics are detailed in both sections 5 and 6 -- we use Titan V100 GPUs and NCCL2.4.7 with Infiniband interconnect.
>
> As described in Section 5, DD-PPO introduces a single additional hyperparameter, the preemption threshold.  We study this hyper-parameter in section 5 -- figure 4 on page 5 shows its effect and figure 6 on page 13 provides a further breakdown.  We find that under values of 60% and 80%, DD-PPO scales near-linearly under both heterogenous and homogenous workloads.
>
> DD-PPO also inherits all of PPO’s hyperparameters, we specified these in section 4 (page 5 under the “Training.” heading).
>
> > - Eq.1: expectation should also be w.r.t transition's stochasticity.
>
> The $\tau \sim \pi$ notation takes into account transition stochasticity.  The trajectory $\tau$ is constructed by sampling actions from $\pi$ and sampling the next state from $\mathcal{T}(s_t, a_t)$.  We have clarified this notation.
>
> > The Transfer Learning tasks is a general setting for for any methods, which are not limited to only Distributed Approach. The results and setting there do not have links such as why and how distributed approaches help such transfer learning.
>
> As we describe in our paper, there is 2-hop connection from our distributed approach (DD-PPO) to transfer learning experiments. Using DD-PPO, we are able to train agents for longer and achieve state-of-the-art performance on PointGoal Navigation. Next, the transfer tasks are designed to show that the models learned on PointGoal Navigation can be transferred (in a variety of ways) to other navigation tasks and enable more sample efficient learning than using ImageNet pretrained weights or learning them from scratch. Thus, the transfer experiments are not testing the scaling of DD-PPO, but rather the generality of a model learned via DD-PPO. R2 notes that "The transfer tasks, to the 'flee' and 'exploration' tasks, were interesting, and further reinforced the results in the paper."

---

> > ### Author Response · Authors · 2019-11-14
> > **Response to R1 (part 2 of 2)**
> >
> > Stepping back, R1's review appears to have missed several of our contributions (listed in Section 1 of our paper). Just to make sure we are all on the same page, we restate them for completeness:
> > We present Decentralized Distributed Proximal Policy Optimization (DD-PPO), a method for distributed reinforcement learning in resource-intensive simulated environments. DD-PPO is distributed (uses multiple machines), decentralized (lacks a centralized server), and synchronous (no computation is ever ‘stale’), making it conceptually simple and easy to implement (noted by R1, R2, and R3)
> > In our experiments on training virtual robots to navigate in Habitat-Sim, DD-PPO exhibits near-linear scaling – achieving a speedup of 107x on 128 GPUs over a serial implementation (noted by R1, R2, and R3)
> > We provide an answer to the scientific question: what are the fundamental limits of learnability in PointGoalNav with GPS+Compass? Is this task entirely learnable? We answer this question affirmatively via an ‘existence proof’  (noted by R2 and R3)
> > We achieve state-of-the-art performance on PointGoal Navigation (R2) and study the task itself (R2, R3)
> > We demonstrate that the scene understanding and navigation policies learned on PointGoal Navigation can be transferred to other tasks, indicating that the scene understanding and navigation policies learned can be transferred to other navigation tasks – the analog of ‘ImageNet pre-training + task-specific fine-tuning’ for embodied AI (noted by R2)

---

### Decision · Program_Chairs · 2019-12-19

**Decision:**

Accept (Poster)

**Comment:**

The authors present and implement a synchronous, distributed RL called Decentralized Distributed Proximal Policy Optimization. The proposed technique was validated for pointgoal visual navigation task on recently introduced Habitat challenge 2019 and got the state of art performance.

Two reviews recommend this paper for acceptance with only some minor comments, such as revising the title. The Blind Review #2 has several major concerns about the implementation details. In the rebuttal, the authors provided the source code to make the results reproducible.

Overall, the paper is well written with promising experimental results. I also recommend it for acceptance.